# Insight into Glyproline Peptides’ Activity through the Modulation of the Inflammatory and Neurosignaling Genetic Response Following Cerebral Ischemia–Reperfusion

**DOI:** 10.3390/genes13122380

**Published:** 2022-12-16

**Authors:** Vasily V. Stavchansky, Ivan B. Filippenkov, Julia A. Remizova, Alina E. Denisova, Ivan V. Mozgovoy, Leonid V. Gubsky, Nikolay F. Myasoedov, Lyudmila A. Andreeva, Svetlana A. Limborska, Lyudmila V. Dergunova

**Affiliations:** 1Institute of Molecular Genetics of National Research Center “Kurchatov Institute”, Kurchatov Sq. 2, Moscow 123182, Russia; 2Department of Neurology, Neurosurgery and Medical Genetics, Pirogov Russian National Research Medical University, Ostrovitianov Str. 1, Moscow 117997, Russia; 3Federal Center for the Brain and Neurotechnologies, Federal Biomedical Agency, Ostrovitianov Str. 1, Building 10, Moscow 117997, Russia

**Keywords:** tMCAO, Semax, PGP, PGPL, inflammation, neurotransmission, gene expression, gene networks

## Abstract

Glyprolines are Gly-Pro (GP)- or Pro-Gly (PG)-containing biogenic peptides. These peptides can act as neutrophil chemoattractants, or atheroprotective, anticoagulant, and neuroprotective agents. The Pro-Gly-Pro (PGP) tripeptide is an active factor of resistance to the biodegradation of peptide drugs. The synthetic Semax peptide, which includes Met-Glu-His-Phe (MEHF) fragments of adrenocorticotropic hormone and the C-terminal tripeptide PGP, serves as a neuroprotective drug for the treatment of ischemic stroke. Previously, we revealed that Semax mostly prevented the disruption of the gene expression pattern 24 h after a transient middle cerebral artery occlusion (tMCAO) in a rat brain model. The genes of this pattern were grouped into an inflammatory cluster (IC) and a neurotransmitter cluster (NC). Here, using real-time RT-PCR, the effect of other PGP-containing peptides, PGP and Pro-Gly-Pro-Leu (PGPL), on the expression of a number of genes in the IC and NC was studied 24 h after tMCAO. Both the PGP and PGPL peptides showed Semax-unlike effects, predominantly without changing gene expression 24 h after tMCAO. Moreover, there were IC genes (*iL1b*, *iL6*, and *Socs3*) for PGP, as well as IC (*iL6*, *Ccl3*, *Socs3*, and *Fos*) and NC genes (*Cplx2*, *Neurod6*, and *Ptk2b*) for PGPL, that significantly changed in expression levels after peptide administration compared to Semax treatment under tMCAO conditions. Furthermore, gene enrichment analysis was carried out, and a regulatory gene network was constructed. Thus, the spectra of the common and unique effects of the PGP, PGPL, and Semax peptides under ischemia–reperfusion were distinguished.

## 1. Introduction

Glyprolines are a class of oligopeptides that include an amino acid sequence consisting of Gly-Pro (GP) or Pro-Gly (PG) residues [1]. Glyprolines are natural compounds that are produced by the intra- and extracellular catabolism of collagen, elastin, and related proteins, as well as the proteolysis of dietary proteins. These peptides can act as neutrophil chemoattractants, or atheroprotective, anticoagulant, and neuroprotective agents [2,3,4,5]. The structures of glyprolines result in high metabolic stability. The tripeptide Pro-Gly-Pro (PGP) plays a stabilizing role in peptides, protecting the molecules from premature biodegradation. Furthermore, PGP is used in the creation of synthetic peptides [6,7]. Among the glyprolines, the synthetic Semax (Met-Glu-His-Phe-Pro-Gly-Pro, MEHFPGP) peptide is one of the most studied and is used as a neuroprotective drug for the treatment of ischemic stroke [8,9,10,11,12,13,14]. It should be noted that Semax is a hybrid molecule in which the Met-Glu-His-Phe (MEHF) fragments of adrenocorticotropic hormone (ACTH) have been fused to the C-terminal amino acid sequence PGP. Previously, we studied the effects of Semax on the transcriptome under experimental cerebral ischemia [15,16]. Ischemia was induced by rat transient middle cerebral artery occlusion (tMCAO), which reflects ischemia–reperfusion (IR) processes in humans after strokes. We revealed that Semax prevented disorders in the expression of genes associated with the inflammatory and neurotransmitter responses of the ischemic brain tissue of model animals [15,16]. Moreover, the correction of the activity of key proteins involved in inflammation and cell-death processes (MMP-9, c-Fos, and JNK), as well as neuroprotection and recovery (CREB), may contribute to the neuroprotective action of Semax in IR conditions [17]. However, the contribution of Semax’s structural parts, particularly its glyproline unit, to the neuroprotective properties remains unclear.

Previously, we showed that both Semax and PGP modulated the expression of genes associated with processes of proliferation, differentiation, migration, survival, and cell death in a permanent middle cerebral artery occlusion (pMCAO) rat model. This model reflects the processes after acute stroke [18,19]. It was also shown that PGP is the most stable decay product of Semax, and therefore, its effects may contribute to Semax’s action [20]. Concomitantly, the effect of PGP on the expression of growth factor genes only partially coincided with the effect of Semax after pMCAO [21]. Under the influence of PGP, the expression of genes associated with the function of the immune response was significantly reduced, the opposite to the effect of Semax under permanent cerebral ischemia conditions [19,22,23]. Therefore, PGP’s impact on the transcriptome may be similar to Semax’s in some aspects, but PGP’s involvement requires further study. In particular, a comparison of the effects of different PGP-containing peptides, using transcriptome analysis and other model systems, may be relevant.

In this study, we used PGP and another synthetic PGP-containing peptide, Pro-Gly-Pro-Leu (PGPL), in the rat tMCAO model. It should be noted that the PGP and PGPL peptides have numerous effects, which may be beneficial for the treatment of cerebrovascular accidents [24]. The multiple intranasal administration of PGP and PGPL to rats with hypercholesterolemia and increased blood clotting led to the normalization of cholesterol levels and the restoration of impaired functions of the anticoagulant system [24,25,26]. Under tMCAO conditions, we studied the effects of PGP and PGPL on the expression of a number of inflammatory cluster (IC) and neurotransmitter cluster (NC) genes in the rat brain and compared the effects of peptides with those of Semax using real-time RT-PCR. Furthermore, gene enrichment analysis was carried out, and a regulatory gene network was constructed. Thus, the spectra of the common and unique effects of the PGP, PGPL, and Semax peptides under IR were determined.

## 2. Materials and Methods

### 2.1. Animals

White two-month-old male rats of the Wistar line (weight: 200–250 g) were obtained from the AlCondi, Ltd., animal breeding house, Moscow, Russian Federation. The animals were divided into five groups: “sham operation” (SH), “ischemia–reperfusion” (IR), “ischemia–reperfusion + Semax” (IS), “ischemia–reperfusion + PGP” (IP), and “ischemia–reperfusion + PGPL” (IL). Each experimental group included at least eight animals.

### 2.2. Rat Transient Middle Cerebral Artery Occlusion Model

#### 2.2.1. Operation

The transient middle cerebral artery occlusion (tMCAO) model was induced by endovascular occlusion of the right middle cerebral artery using a monofilament (Doccol Corporation, Sharon, MA, USA) for 90 min and was performed under magnetic resonance imaging (MRI), as described previously [27]. The sham-operated rats (group “SH”) were subjected to a similar surgical procedure under anesthesia (neck incision and separation of the bifurcation), but without tMCAO. The rats were decapitated at 24 h after tMCAO/sham operations.

#### 2.2.2. Peptide Administration

To the rats of the IS, IP, and IL groups, the peptides Semax, PGP, and PGPl were administered intraperitoneally at doses of 100, 37.5, and 200 µg/kg rat weight, respectively. Animals of IR and SH groups were injected with saline. Peptides or saline were administered intraperitoneally after 1.5 h, as well as at 2.5 and 6.5 h after the surgical procedure. The used concentrations of substances and the time of peptide administration correspond to the data from the literature [15,19,22,23,28,29,30].

### 2.3. Sample Collection and RNA Isolation

The ipsilateral subcortical structures were obtained, and total RNA was isolated, as previously described [31]. The RNA integrity was checked by analyzing the ratio of the bands of 28S and 18S rRNAs after denaturing agarose gel electrophoresis [32].

### 2.4. cDNA Synthesis and Real-Time Reverse Transcription Polymerase Chain Reaction (RT-PCR)

cDNA using oligo (dT)_18_ primers was synthesized, and RT-PCR was conducted as previously described [31]. PCR-primers were selected using the OLIGO Primer Analysis Software version 6.31 and were synthesized by the Evrogen Joint Stock Company (Appendix A).

### 2.5. Data Analysis of Real-Time RT-PCR and Statistics

The reference gene *Gapdh* was used to normalize the expression for the cDNA samples. The relative gene expression was calculated by the 2^−ΔΔCt^ method [33]. The data were analyzed using the REST gene quantification program (Freising-Weihenstephan, Germany); the significance of the differences in expression in the control and experimental groups was assessed using the randomization criterion [34]. At least eight animals were included in each comparison group. When comparing data groups, statistically significant differences were considered with the probability *p* < 0.05, as previously described [31].

### 2.6. Functional Analysis

The Database for Annotation, Visualization and Integrated Discovery (DAVID) v.6.8. [35], Gene Set Enrichment Analysis (GSEA) [36], and gProfileR [37] were used to annotate the functions of the differentially expressed mRNAs (DEGs). Only functional annotations that had *p*-values, adjusted using the Benjamini–Hochberg procedure, lower than 0.05 (*Padj* < 0.05) were considered. Principal component analysis (PCA) of normalized read counts was conducted using R software. Hierarchical cluster analysis of DEGs was performed using Heatmapper (Wishart Research Group, University of Alberta, Ottawa, ON, Canada) [38]. A volcano plot was constructed using Microsoft Excel. The Cytoscape 3.8.2 software (Institute for Systems Biology, Seattle, WA, USA) [39] was used to visualize the regulatory network. Additional calculations were performed using Microsoft Excel.

## 3. Results

### 3.1. Magnetic Resonance Imaging (MRI)

We detected the locations and volumes of ischemic foci in animals 24 h after tMCAO using the diffusion-weighted imaging (DWI) and T2-weighted imaging (T2 WI) modes of MRI. All ischemic rats from any of IR, IS, IP, and IL groups had ischemic injury in right hemisphere of the brain. A typical DWI with an ADC map and T2 WI scans of the formation of ischemic injury areas with a subcortical localization in the brains of rats 24 h after tMCAO are shown in Figure 1.

### 3.2. Selection of Genes for Studying PGP-Containing Peptides’ Action under Ischemia–Reperfusion (IR) Conditions Based on Our Previous Results

In this study, transcriptomic data obtained previously in [15] were subjected by principal component analysis (PCA). Therefore, the mRNA transcriptomes of the three experimental groups (SH, IR, and IS) were compared using PCA. Appendix A illustrates a clear separation between all groups. Then, six genes (*iL1b*, *iL6*, *Ccl3*, *Socs3*, *Hspb1*, and *Fos*) from an inflammatory cluster (IC) and six genes (*Cplx2*, *Neurod6*, *Gabra5*, *Chrm1*, *Gria3*, and *Ptk2b*) from a neurotransmitter cluster (NC) associated with Semax’s action under IR conditions 24 h after tMCAO were selected for analysis based on our previous data [15,16]. Volcano plots illustrate the differences in mRNA expression of a number of selected genes (fold change ≥ 1.5; *Padj* < 0.05) between the IS and IR groups as determined by RNA-seq (Appendix A) and reverse transcription polymerase chain reaction (RT-PCR) (Appendix A).

The real-time RT-PCR analysis of the expression of all the selected genes for IS versus IR is shown in Figure 2a. All of them were verified as DEGs in this pairwise comparison group. Moreover, these genes were DEGs in subcortical structures under IR versus new brain samples of sham-operated rats (SH). Therefore, Semax was associated with compensation for the mRNA expression patterns that were disrupted under IR conditions for both IC and NC clustered genes (Figure 2b). The bar plot illustrates that IR alone activated the expression of IC genes and suppressed the expression of NC genes, in contrast to Semax’s effects 24 h after tMCAO (Figure 2b).

### 3.3. Analysis of the Effects of PGP and PGPL on the Expression of the IC and NC Genes versus IR’s Effect 24 h after tMCAO

Using real-time RT-PCR, the expression changes for the IC (*iL1b*, *iL6*, *Ccl3*, *Socs3*, *Hspb1*, and *Fos*) and NC (*Cplx2*, *Neurod6*, *Gabra5*, *Chrm1*, *Gria3*, and *Ptk2b*) genes under the influence of PGP and PGPL in the subcortical structures of the rat brain 24 h after tMCAO were studied.

Figure 2c illustrates the differential expression results associated with PGP’s action in IP versus IR. We revealed that PGP administration did not induce any significant changes in the expression of the IC genes. Concomitantly, there was only one DEG, *Gabra5*, among the NC genes. Its expression was significantly upregulated by 1.72-fold (*p* = 0.004) in IP versus IR (Figure 2c).

The differential expression of the IC and NC genes associated with PGPL’s action under IR conditions is illustrated in Figure 2d. The administration of PGPL did not produce significant changes in the mRNA levels for IC genes versus IR. However, three NC genes were differentially expressed in IL versus IR. The upregulation of the *Cplx2* (fold change = 2.17, *p* ≤ 0.001), *Chrm1* (fold change = 1.85, *p* = 0.011), and *Gria3* (fold change = 1.70, *p* = 0.017) genes under PGPL treatment 24 h after tMCAO was observed (Figure 2d).

### 3.4. A Search for DEGs Associated with Semax Treatment Compared to PGP and PGPL Treatment under tMCAO Conditions

We searched for DEGs that were associated with Semax treatment, compared to PGP and PGPL treatment, under tMCAO conditions. The bar plots in Figure 2e,f illustrate the expression data for each studied gene in the IS-versus-IP and IS-versus-IL pairwise comparisons, respectively.

We revealed that Semax administration did not lead to statistically significant changes in the expression of the NC genes relative to the levels of expression of these genes in ischemic rats after PGP administration (Figure 2e). However, three IC genes were DEGs in IS versus IP. The downregulation of the *Socs3* (fold change = 0.39, *p* = 0.040), *iL1b* (fold change = 0.32, *p* = 0.027), and *iL6* (fold change = 0.39, *p* = 0.034) genes under Semax versus PGP treatment 24 h after tMCAO was observed (Figure 2e).

Semax treatment was associated with four DEGs (*iL6*, *Ccl3*, *Socs3*, and *Fos*) from IC and three DEGs (*Cplx2*, *Neurod6*, and *Ptk2b*) from the NC related to PGPL treatment (Figure 2f). The upregulation of *Neurod6* (fold change = 2.42, *p* = 0.010) and *Ptk2b* (fold change = 1.32, *p* = 0.038) and downregulation of *Cplx2* (fold change = 0.69, *p* = 0.021), as well as *Socs3* (fold change = 0.52, *p* = 0.018), *Fos* (fold change = 0.43, *p* = 0.005), *iL6* (fold change = 0.50, *p* = 0.013), and *Ccl3* (fold change = 0.43, *p* = 0.018), were observed in IL versus IS 24 h after tMCAO (Figure 2f). The hierarchical cluster analysis of the gene expression profiles obtained in different pairwise comparisons is illustrated in Figure 2g. The differential expression profile for these comparison groups reflects the preservation of two gene clusters. Within these clusters, the effects of the Semax, PGP, and PGPL peptides at the gene expression level under IR conditions were revealed (Figure 2g).

### 3.5. Update of Signaling Pathways Associated with Semax Treatment Based on the Combination of RNA-Seq and PCR Results under the tMCAO Conditions

The combination of RNA-seq and RT-PCR results obtained in this study and in previous studies [15,16] allowed us to update the signaling pathway annotations for Semax treatment under tMCAO conditions. The list included 400 DEGs in IS versus IR. Using David v6.8, 43 signaling pathways with *Padj* < 0.05 were revealed. Among them, 18 signaling pathways were newly annotated, whereas 25 pathways overlapped with the previously annotated pathways in [15], revealed based on the RNA-seq data (Figure 3a). Here, we found signaling pathways (leishmaniasis, malaria, inflammatory bowel disease (IBD), chemokine, TNF, and others) that are involved in inflammation, predominantly under Semax treatment, 24 h after tMCAO. Moreover, they were mainly associated with the downregulated DEGs in IS versus IR (Figure 3b). Additional evidence to validate our findings was obtained using GSEA (Appendix A) and gProfileR (Appendix A) tools.

### 3.6. The Search for Signaling Pathways That Reflect Common and Unique Gene Expression Effects of PGP, PGPL, and Semax Peptides in IR Conditions

Among the 43 signaling pathways identified under IR and Semax treatment, we selected those that were associated with at least one of the studied genes from the NC and IC based on David and KEGG annotation data. Thereby, 27 such signaling pathways were selected. Figure 4 shows a network involving 12 genes in the presentation of 27 signaling pathways associated with Semax treatment 24 h after tMCAO. The nodes designate the genes or signaling pathways in the network. Each line connecting the nodes indicates the involvement of a gene in the functioning of a particular signaling pathway. In a network, all the signaling pathways formed three clusters: a Semax-unlike cluster for both the PGP and PGPL peptides, a Semax-unlike cluster for the PGPL but not PGP peptide, and a Semax-like cluster for both the PGP and PGPL peptides (Figure 4). 

The first pathway cluster included the largest number of signaling pathways, 15. Such pathways were associated with genes that were DEGs in both the IS-versus-IP and IS-versus-IL pairwise comparisons. For instance, the TNF signaling pathway had the highest number of connecting lines with DEGs in IS versus IP and IS versus IL. In particular, the *Socs3*, *Fos*, *Ccl3*, and *iL6* genes were DEGs versus PGP, whereas the *iL6*, *il1b*, and *Socs3* genes were DEGs versus PGPL under Semax treatment 24 h after tMCAO. Therefore, the first pathway cluster can reflect Semax-unlike effects for both the PGP and PGPL peptides. It should be noted that all the studied genes involved in the cluster functioning belonged to the IC. Most of the pathways of this cluster (TNF, PI3K, MAPK, malaria, amoebiasis, and others) predominantly translated signals of inflammatory and immune responses (Figure 4).

The second pathway cluster included eight signaling pathways, which were associated with genes that were DEGs in IS versus IL, but non-DEGs in IS versus IP. Thus, the chemokine signaling pathway had the highest number of connecting lines with DEGs in IS versus IL, namely, *Ptk2b* and *Ccl3*. Concomitantly, these genes were non-DEGs in IS versus IP. Therefore, the second pathway cluster can reflect Semax-unlike effects for PGPL but not PGP peptides. It should be noted that two genes (*Ccl3* and *Fos*) involved in cluster functioning belonged to the IC. However, the majority of the genes (three) belonged to the NC. Thus, the *Ptk2b* gene was downregulated in IS versus IL, and *Chrm1* and *Gria3* were non-DEGs in both pairwise comparisons with the NC. Such genes were associated with neurosignaling pathways (cAMP, calcium, dopaminergic synapse, and others) that were included in the second pathway cluster (Figure 4).

The third pathway cluster included four signaling pathways. They were associated with genes that were non-DEGs in both the IS-versus-IP and IS-versus-IL pairwise comparisons. These were morphine addiction, glutamatergic synapse, retrograde endocannabinoid signaling, and neuroactive ligand–receptor interaction pathways that predominantly translated signals from the central nervous system (CNS). The *Gabra5*, *Chrm1*, and *Gria3* genes, which were non-DEGs in both IS versus IP and IS versus IL, had connecting lines to these pathways. The change in the expression profile of all the associated genes between Semax and PGP and PGPL treatment did not differ significantly. Thus, the third pathway cluster can reflect Semax-like effects for both PGPL and PGPL peptides (Figure 4).

Thus, a regulatory network illustrated the spectra of the effects of the Semax, PGP, and PGPL peptides on gene expression in the rat brain under IR conditions, against the background of the structures having a common PGP unit.

## 4. Discussion

In this study, we used three PGP-containing peptides (Semax, PGP, and PGPL) to differentiate their effects in a rat tMCAO model. The Semax peptide combines the ACTH(4-7) and PGP structures and probably integrates the properties of each of the subunits. Semax is one of the most studied and practically significant peptides used in stroke therapy. The glyprolines PGP and PGPL can also be considered potential agents for the prevention and treatment of pathologies. They are able to exhibit neuroprotective, antihypertensive, anti-inflammatory, and antithrombotic effects [7,40]. The multiple intranasal administration of PGP and PGPL to rats with hypercholesterolemia and increased blood clotting led to the normalization of cholesterol levels and the restoration of impaired functions of the anticoagulant system [24,25,26]. However, their contribution to gene expression during ischemia is not known for certain.

Here, under MRI, a rat tMCAO model was implemented to test the peptides’ effects. Molecular genetics instruments were used. Six inflammatory (*iL1b*, *iL6*, *Ccl3*, *Socs3*, *Hspb1*, and *Fos*) and six neurotransmitter (*Cplx2*, *Neurod6*, *Gabra5*, *Chrm1*, *Gria3*, and *Ptk2b*) genes associated with Semax treatment under IR conditions were selected based on our previous data [15,16]. Using real-time RT-PCR, we confirmed changes in gene expression under conditions of ischemia and Semax administration in the subcortical structures of model rats 24 h after tMCAO. Additionally, the hierarchical clustering of gene expression data provided two clusters of genes of inflammation (IC) and neurotransmission (NC), respectively. Moreover, the genes of NC were downregulated after tMCAO, whereas they were upregulated after Semax administration. On the other hand, the IC genes were upregulated after tMCAO, whereas they were downregulated after Semax administration. It should be noted that the studied NC genes were associated with the processes of intercellular signal transduction, excitotoxic damage, and neuroprotection of nerve cells under IR conditions [41,42,43,44,45,46,47]. The proteins encoded by the IC genes are involved in a range of processes that control the inflammatory response [48,49,50,51,52,53,54]. Thus, the studied genes are thought to play a role as potential targets of stroke therapy or to participate in the implementation of protective reactions induced by drugs. Taking together our previous ([15,16]) and present results, we expanded the range of signaling pathways associated with Semax. Most of the newfound signaling pathways have been associated with the formation of inflammatory and immune responses.

Then, changes in the expression of the IC and NC genes under treatment with short PGP-containing peptides (PGP and PGPL) in the subcortical brain structures of rats 24 h after tMCAO were studied. Both the PGP and PGPL peptides showed effects, predominantly without changes in the gene expression under IR. This result emphasizes the importance of the structure of ACTH (MEHF) for Semax’s effects. In particular, the administration of the PGP and PGPL peptides did not have a statistically significant effect on the expression of the IC genes. It was shown that the PGP fragment structurally reproduces the key unit of the CXCL2 chemokine. CXCL2 is responsible for the activation of neutrophils. Moreover, PGP is able to bind to the CXCR2 receptor and partially recapitulate the chemoattractant function for neutrophils [55,56].

Among the NC genes, we found an increase in expression only for the *Gabra5* gene, encoding the α5 subunit of the γ-aminobutyric acid (GABA) receptor after PGP administration 24 h after tMCAO. The regulation of the activity of the GABA receptor is important in the formation of a protective reaction in response to the processes of excitotoxicity and microglia activation during ischemia [42,57,58]. Additionally, three genes (*Cplx2*, *Chrm1*, and *Gria3*) were increased in expression upon PGPL administration to ischemic rats. Such genes belong to the NC, and the proteins encoded by these genes are involved in the regulation of neurotransmitter processes. Moreover, the repression of the neurotransmitter system under conditions of cerebral ischemia has been noted previously [41,45,59,60]. Our data did not exclude the role of PGPL in the correction of neurotransmission activity impaired by IR.

A search for DEGs associated with Semax treatment, compared to PGP and PGPL treatment, under tMCAO conditions was carried out. The results obtained made it possible to identify genes that characterize the overlapping and unique molecular signatures of the peptides’ effects under IR conditions. Both the PGP and PGPL peptides showed Semax-unlike effects, predominantly without changing gene expression 24 h after tMCAO. Moreover, there were IC genes (*iL1b*, *iL6*, and *Socs3*) for PGP, as well as IC (*iL6*, *Ccl3*, *Socs3*, and *Fos*) and NC genes (*Cplx2*, *Neurod6*, and *Ptk2b*) for PGPL, that significantly changed in expression levels after peptide administration compared to Semax treatment under tMCAO conditions. Based on the differential expression results, a functional network was constructed. It linked genes from the IC and NC with 27 signaling pathways associated with Semax treatment during IR. At the same time, we showed that not all the Semax-related signaling pathways are equally characteristic of the effects of short PGP-containing PGP and PGPL peptides. The structures of PGP and PGPL are only partially similar to the structure of Semax, and the differences are great. Therefore, the network can specify the relationship between peptide structures and their functional effects.

In the network, the signaling pathways formed three clusters depending on the effects of the peptides on gene expression. The first pathway cluster included 15 signaling pathways that were associated with genes that were differentially expressed after Semax treatment, versus both PGP and PGPL, under IR conditions. Such a pathway cluster can reflect Semax-unlike effects for both PGP and PGPL peptides. Interestingly, only the IC genes served as nodes in the first pathway cluster. In particular, the *Socs3*, *Fos*, *Ccl3*, and *iL6* genes were DEGs versus PGP, whereas the *iL6*, *il1b*, and *Socs3* genes were DEGs versus PGPL under Semax treatment 24 h after tMCAO. Therefore, the first pathway cluster involved a predominantly inflammatory and immune response, modulated by short PGP-containing peptides in a manner different to that of Semax. The short PGP and PGPL peptides probably do not have such a pronounced effect on the inflammatory and immune genetic responses under IR. According to our data, this is evidenced by the absence of significant downregulated IC genes under their influence relative to the effect of saline 24 h after tMCAO. It can be assumed that the more powerful Semax-related downregulation of the IC genes, and the Semax effect corresponding to the first pathway cluster, were mostly elicited by the ACTH(4-7) fragment, which is absent in the PGP and PGPL peptides. The manifestation of ACTH(4-7)’s action may be associated with the so-called synactonic mechanism. Synactone can be defined as a combination of the peptide itself and the products of its metabolism, acting in a certain sequence and by interaction [61,62,63,64]. Both the main parent peptide and secondary peptides of its metabolism have their own binding sites and together make up a single complex of bioregulators [61,62,63,64]. The secondary metabolites of Semax are known, and their diversity is ensured by the active metabolism of the ACTH(4-7) fragment [61]. Moreover, in addition to orthosteric binding, allosteric interactions of peptides with various types of receptors have been observed, indicating the multiple (pleiotropic) nature of their actions [61,62,63,64].

The second pathway cluster included eight signaling pathways. They were associated with genes that were differentially expressed under IR conditions after Semax treatment versus PGPL, but not versus PGP administration. This cluster can reflect the Semax-unlike effect of PGPL, but not that of PGP peptides. Such a cluster included signaling pathways mainly related to the processes of neurosignaling and neuroreception. Indeed, PGP administration does not lead to statistically significant changes in the expression of NC genes relative to the level of expression of these genes in ischemic rats after Semax administration. Concomitantly, Semax treatment was associated with three DEGs (*Cplx2*, *Neurod6*, and *Ptk2b*) from the NC related to PGPL treatment. Notably, the *Fos* and *Ptk2b* genes demonstrated a Semax-unlike expression pattern after PGPL administration. Such genes were associated with the inflammatory component of the second signaling cluster. The *Fos* gene encodes a transcriptional factor that is involved in the activation of the expression of a number of interleukin and chemokine genes during ischemia [52]. The tyrosine protein kinase PTK2b functions in signaling from cytokine receptors, growth factors, and G-protein-coupled receptors. The excessive activation of the PTK2b protein may contribute to neurotoxic effects [65]. It should be noted that the differences in the functioning of the signaling pathways leading to the formation of the second pathway cluster may be associated with the presence of C-terminal leucine in PGPL molecule [7]. Thus, Semax- and PGP-unlike patterns of metabolites (synactone) can form during the proteolysis of PGPL peptides [66,67]. Therefore, a unique spectrum of PGPL effects can be observed under IR conditions. The effects of the PGPL peptide and its metabolites in the nervous cells are not fully understood, although specific PGPL-binding sites on the cytoplasmic membrane of the basal ganglia of the rat brain are known [24]. In addition, PGPL peptide exhibits hypocholesterolemic, anticoagulant, and fibrinolytic effects that are more pronounced than those of the PGP and Semax peptides [40].

The third pathway cluster included four signaling pathways that were associated only with non-DEGs after Semax treatment, versus both PGP and PGPL treatment, under IR conditions. The expression profiles of all the associated genes between Semax and PGP and PGPL treatment did not differ. Thus, the third pathway cluster can reflect Semax-like effects for both the PGPL and PGPL peptides. The genes included in the signaling pathways of this cluster belong to NC (*Chrm1*, *Gria3*, and *Gabra5*) and encode acetylcholine, glutamate, and GABA receptors. The appearance of the third cluster of signaling pathways may be due to the presence of a common PGP structural unit in the structures of all the studied peptides. It is known that PGP is the main metabolite of Semax and PGPL and represents the common functional core of synactone for these peptides [61,66,68]. Moreover, PGP is capable of both orthosteric [69,70] and allosteric binding to the receptors [61,63,68]. The ability of PGP to modulate the activity of acetylcholine, glutamate, and other receptor systems involved in neurosignaling processes was shown [61,63,68].

Thus, our data provide insight into glyproline peptides’ activity through the modulation of the inflammatory and neurosignaling genetic responses following cerebral IR. The main limitation of our study is the lack of genome-wide expression data after the administration of the PGP and PGPL peptides under tMCAO conditions. A detailed study using RNA-seq will more accurately identify differences in expression patterns caused by synthetic peptides and their units. The results will also underlie the selection of targets for analysis of peptide functioning in protein levels in order to get even closer to understanding the phenomenon of their properties.

## 5. Conclusions

In conclusion, the comparison of glyproline peptides allowed us to determine their general and individual effects on gene expression under cerebral IR. Furthermore, the results served as a first step in understanding the relationship between the chemical structure of the glyproline peptides and their possible effects at the genomic and functional levels. We believe that understanding the genesis of the relationship between the structure and function of peptides can lead to the creation of drugs that are even more effective than those currently available.

## Figures and Tables

**Figure 1 genes-13-02380-f001:**
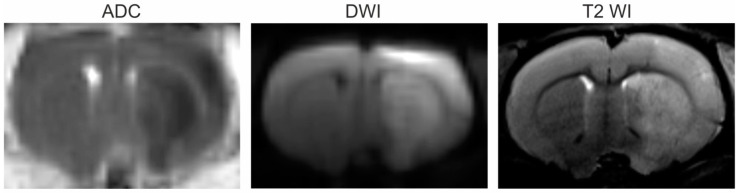
MRI of ischemic foci 24 h after tMCAO. DWI with an ADC map and T2 WI scans show a subcortical localization of ischemic injury in the ipsilateral cerebral hemispheres in rats after tMCAO.

**Figure 2 genes-13-02380-f002:**
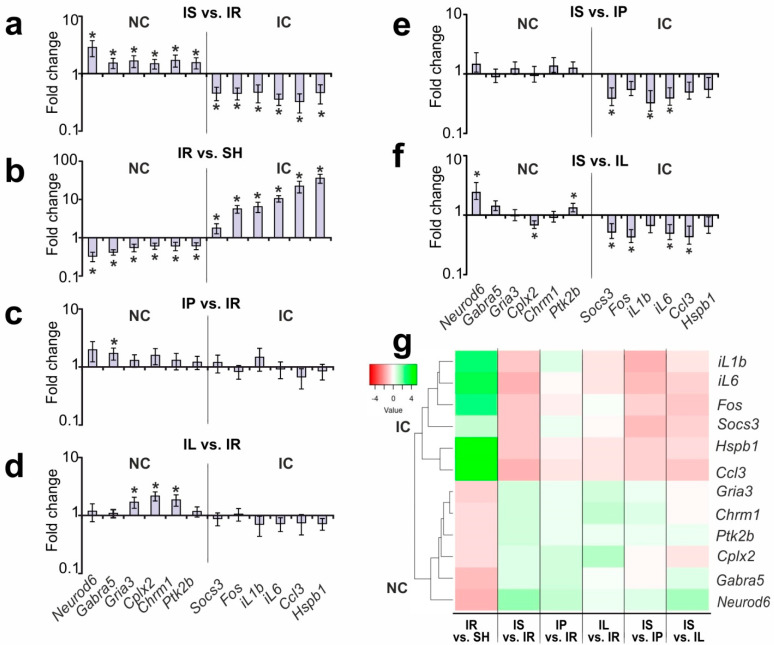
RT-PCR analysis of DEGs and discrimination of peptides’ effects under IR conditions (24 h after tMCAO). (**a**–**f**) Analysis of the changes in the gene expression of IC and NC clusters in IS versus IR (**a**), IR versus SH (**b**), IP versus IR (**c**), IL versus IR (**d**), IS versus IP (**e**), and IS versus IL (**f**). The data are presented as the means ± standard errors of the means (SEs). The statistical significance (*p* < 0.05) is indicated above the bars by an asterisk (*). (**g**) Hierarchical cluster analysis of gene expression profiles obtained in different pairwise comparisons. Each column represents a comparison group, and each row represents a DEG. Green strips represent high relative expression, and red strips represent low relative expression, with at least eight per group.

**Figure 3 genes-13-02380-f003:**
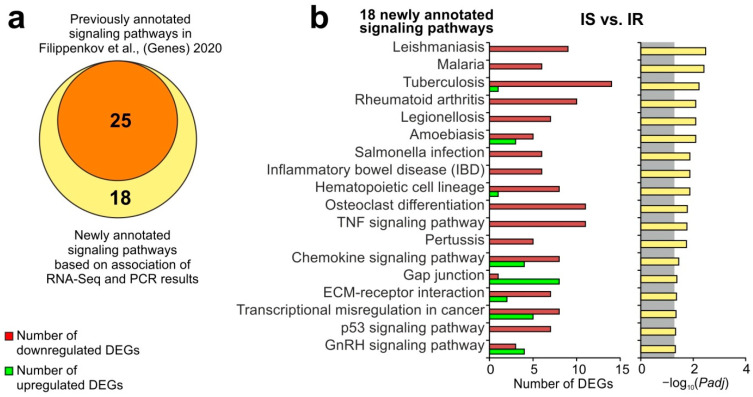
Update of signaling pathways associated with DEGs in IR versus IS based on the combination of RNA-seq and RT-PCR results after 24 h under tMCAO conditions. KEGG database analysis of DEGs was carried out using the DAVID database. (**a**) The number of signaling pathways that overlapped between previous and new enrichment results [15]. (**b**) The number of upregulated and downregulated DEGs, as well as the *p*-values, adjusted using the Benjamini–Hochberg procedure (*Padj*) for 18 newly annotated signaling pathways. Only those genes and signaling pathways where *Padj* < 0.05 were selected for analysis. Those with *Padj* ≥ 0.05 are against a gray background.

**Figure 4 genes-13-02380-f004:**
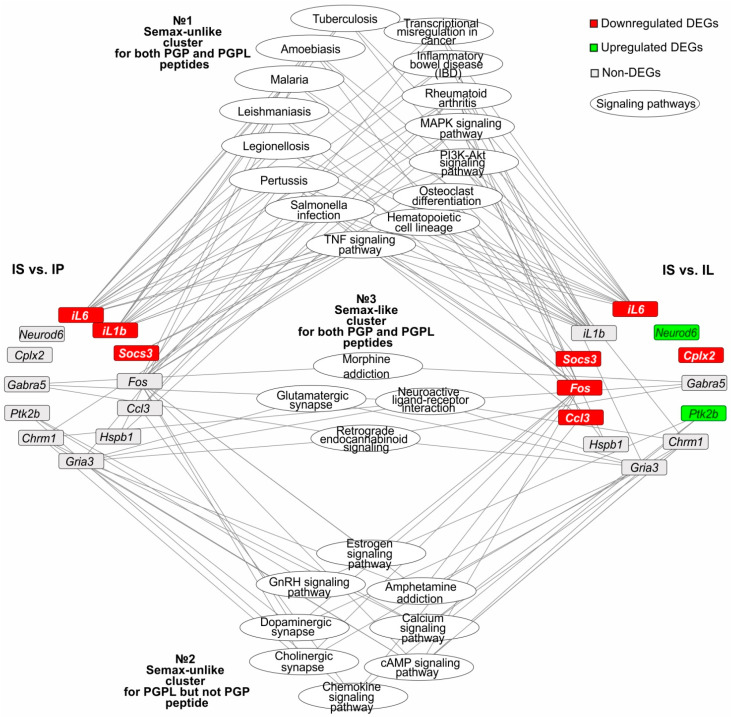
A regulatory network that reflects common and unique gene expression effects of PGP, PGPL, and Semax peptides in IR conditions. The network was constructed using Cytoscape 3.8.2 (Institute for Systems Biology, Seattle, WA, USA). The oval blocks at the nodes represent signaling pathways. The gray, green, and red rectangular boxes represent non-, up- and downregulated genes, respectively. The lines connecting the oval and rectangular blocks indicate the participation of the protein products of genes in the signaling pathway.

## Data Availability

Publicly available datasets were analyzed in this study. These data can be found here: [71].

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
