# Peer review of "Insight into Glyproline Peptides’ Activity through the Modulation of the Inflammatory and Neurosignaling Genetic Response Following Cerebral Ischemia–Reperfusion"

_genes, 2022, doi:10.3390/genes13122380_

Round 1

Reviewer 1 Report

This manuscript demonstrates that the protective effects of the PGP and PGPL like-semax peptide on cerebral ischemia reperfusion by transcriptome method, and the results show that there were IC genes (iL1b, iL6, and Socs3) for PGP, as well as IC (iL6, Ccl3, Socs3, and Fos) and NC genes (Cplx2, Neurod6, and Ptk2b) for PGPL, that significantly changed in expression levels after peptide administration under tMCAO conditions.

1.This manuscript is based on the study of Semax's protection against cerebral ischemia and reperfusion to observe the differences in the protection mechanisms of the Semax-like peptide PGP and PGPL, so the similarities and differences in the neuroprotective mechanisms between the three peptide need conduct detailed analysis in the discussion section, such as changes in those expression differences.

2.This manuscript uses transcriptome and real-time RT-PCR methods to prove that gene expression in SH, IR, IS, IP, and PGPL group are different at the transcriptional level, but when functioning, changes in protein levels are more likely to affect changes in function, so it is recommended that the authors increase the results of protein level changes.

3. In 2.2.2 section, why are the amounts and timing of intraperitoneal injections of Semax, PGP and PGPL different? In addition, can intraperitoneally injected Semax, PGP, and PGPL across the blood-brain barrier to exert neuroprotective effects?

Author Response

Response to the comments of Reviewer 1 to Manuscript ID: genes-1980509

Authors:

We are very grateful to the Reviewer 1 for the review and constructive comments. We carefully considered the comments of the Reviewer 1 and attached the answers to all comments.

Reviewer 1:

  1. This manuscript is based on the study of Semax's protection against cerebral ischemia and reperfusion to observe the differences in the protection mechanisms of the Semax-like peptide PGP and PGPL, so the similarities and differences in the neuroprotective mechanisms between the three peptide need conduct detailed analysis in the discussion section, such as changes in those expression differences;

Authors:

In accordance with the Reviewer’s recommendation, changes were made (lines 378-383 in Mark-up copy).

Reviewer 1:                                                                                       

  1. This manuscript uses transcriptome and real-time RT-PCR methods to prove that gene expression in SH, IR, IS, IP, and PGPL group are different at the transcriptional level, but when functioning, changes in protein levels are more likely to affect changes in function, so it is recommended that the authors increase the results of protein level changes.

Authors:

We are very grateful to the Reviewer for the comment. However, the protein expression analysis is an independent study and plan to be performed by us in the future. Here, we consider the study of brain protein profile in the groups mentioned by the Reviewer as one of the limitations of our study. In accordance with the Reviewer’s recommendation, changes were added in the text (lines 460-462 in Mark-up copy).

Reviewer 1:

  1. In 2.2.2 section, why are the amounts and timing of intraperitoneal injections of Semax, PGP and PGPL different? In addition, can intraperitoneally injected Semax, PGP, and PGPL across the blood-brain barrier to exert neuroprotective effects?

Authors:

In accordance with the Reviewer’s recommendation, changes were made (lines 106, 108-112 in Mark-up copy).

In addition, in the norm the ability of Semax, PGP, and PGPL peptides to penetrate the blood-brain barrier (BBB) after injection has been shown previously (Potaman et al. 1991 (PMID: 1652713); Ashmarin et al. 2008 (PMID: 18695718); Shevchenko et al. 2014 (PMID: 25403393); Zolotarev et al. 2003 (PMID: 12910309)). It should be noted that the barrier function of the BBB decreases under conditions of ischemia-reperfusion damage. Therefore, the BBB becomes even more permeable to peptides.

Reviewer 2 Report

In this manuscript, the authors sought to explore the effects of Semax and two other peptides on transcriptome data of the brain after tMCAO. The language is clear, but written with too little details. For example, the materials and methods portion lacks many crucial things needed for reproducibility. The point-by-point comments and suggestions are below.

1) In the first section of results, the authors showed three MRI images of the rat brain at 24h after tMCAO. Did the animals undergo treatment yet? How does Semax affect the ischemic foci as seen in MRI?

2) In the second part, the authors compared transcription data between IS and IR, were they all treated with peptides? It is confusing to read the groups. Some of the groups, like IL, didn't have a full name in the text. 

3) What do the samples look like in PCA or PLSDA analysis of the normalized transcription counts? The authors didn't show any overview data for the samples within each group.

4) What does Semax itself do to the brain? I may be mistaken, but I don't see a Sham/Semax group in the data. 

5) Finally, the authors updated the pathways connected to Semax, PGP and PGPL, but what does it mean for genes involved with tuberculosis or malaria to be differentially expressed in stroke? The authors should provide more evidence to validate and explain these findings. 

Author Response

Response to the comments of Reviewer 2 to Manuscript ID: genes-1980509

Authors:

We are very grateful to the Reviewer 2 for the review and constructive comments. We carefully considered the comments of the Reviewer 2 and attached the answers to all comments.

Reviewer 2:

  1. In the first section of results, the authors showed three MRI images of the rat brain at 24h after tMCAO. Did the animals undergo treatment yet? How does Semax affect the ischemic foci as seen in MRI?

Authors:

All ischemic rats from any of IR, IS, IP and IL groups had ischemic injury in right hemisphere of the brain by MRI. In accordance with the Reviewer’s recommendation, changes were added in the text (lines 175-176 in Mark-up copy).

Additionally, as we have shown previously, Semax does not significantly affect the volume of the focus, according to MRI (Filippenkov et al. 2020 (PMID: 32580520)). Thus, a typical DWI with an ADC map and T2 WI scans of the formation of ischemic injury areas with a subcortical localization in the brains of rats 24 h after tMCAO (IR) are shown in Figure 1.

Reviewer 2:

  1. In the second part, the authors compared transcription data between IS and IR, were they all treated with peptides? It is confusing to read the groups. Some of the groups, like IL, didn't have a full name in the text.

Authors:

Abbreviations for all groups were deciphered in the text (lines 93-95 in Mark-up copy).

Reviewer 2:

  1. What do the samples look like in PCA or PLSDA analysis of the normalized transcription counts? The authors didn't show any overview data for the samples within each group

Authors:

In accordance with the Reviewer’s recommendation, PCA plot was added in Figure S1 (Supplementary files). Also, changes were added in the text (lines 164-165, 184-187, 193-194, 471-472 in Mark-up copy).

Reviewer 2:

  1. What does Semax itself do to the brain? I may be mistaken, but I don't see a Sham/Semax group in the data.

Authors:

Indeed, we did not study the effect of Semax on Sham group. We used the Sham group only to identify genes that altered the expression level in the rat brain in response to ischemia-reperfusion (IR versus SH). Previously, we did not detect any DEGs after Semax administration versus saline administration in brain of health rats (Filippenkov et al., Int J Mol Sci. 2021).

Reviewer 2:

  1. Finally, the authors updated the pathways connected to Semax, PGP and PGPL, but what does it mean for genes involved with tuberculosis or malaria to be differentially expressed in stroke? The authors should provide more evidence to validate and explain these findings.

Authors:

The signaling pathways mentioned by the Reviewer, as well as those related to other infections, reflect the contribution of the genes of the inflammatory and immune component in studied responses.

In accordance with the Reviewer’s recommendation, more evidence was made to validate and explain our findings. Thus, GSEA and gProfileR were used for functional annotation to check our results. The results were added in the text (lines 161, 261-262, 474-475) and in Supplementary Tables 2 and 3.

Round 2

Reviewer 1 Report

Well.